# Pan-ROCK and ROCK2 Inhibitors Affect Dexamethasone-Treated 2D- and 3D-Cultured Human Trabecular Meshwork (HTM) Cells in Opposite Manners

**DOI:** 10.3390/molecules26216382

**Published:** 2021-10-22

**Authors:** Megumi Watanabe, Yosuke Ida, Masato Furuhashi, Yuri Tsugeno, Fumihito Hikage, Hiroshi Ohguro

**Affiliations:** 1Departments of Ophthalmology, School of Medicine, Sapporo Medical University, Sapporo 060-8556, Japan; watanabe@sapmed.ac.jp (M.W.); funky.sonic@gmail.com (Y.I.); yuri.tsugeno@gmail.com (Y.T.); fuhika@gmail.com (F.H.); 2Departments of Cardiovascular, Renal and Metabolic Medicine, Sapporo Medical University, Sapporo 060-8556, Japan; furuhasi@sapmed.ac.jp

**Keywords:** 3D spheroid cultures, human trabecular meshwork (HTM), Rho-associated coiled-coil containing protein kinase (ROCK), ripasudil, KD025, dexamethasone

## Abstract

Effects of a pan-ROCK-inhibitor, ripasudil (Rip), and a ROCK2 inhibitor, KD025 on dexamethasone (DEX)-treated human trabecular meshwork (HTM) cells as a model of steroid-induced glaucoma were investigated. In the presence of Rip or KD025, DEX-treated HTM cells were subjected to permeability analysis of 2D monolayer by transendothelial electrical resistance (TEER) and FITC–dextran permeability, physical properties, size and stiffness analysis (3D), and qPCR of extracellular matrix (ECM), and their modulators. DEX resulted in a significant increase in the permeability, as well as a large and stiff 3D spheroid, and those effects were inhibited by Rip. In contrast, KD025 exerted opposite effects on the physical properties (down-sizing and softening). Furthermore, DEX induced several changes of gene expressions of ECM and their modulators were also modulated differently by Rip and KD025. The present findings indicate that Rip and KD025 induced opposite effects toward 2D and 3D cell cultures of DEX-treated HTM cells.

## 1. Introduction

Quite recently, a new type of ocular hypotensive drug, ripasudil hydrochloride hydrate (Rip), a Rho-associated coiled-coil containing protein kinase (ROCK) inhibitor (ROCK-i), has been made available for use in our clinic as an additional option for an anti-glaucoma medication [1,2]. As a possible mechanism for decreasing IOP by inhibiting ROCKs, ubiquitous downstream effector proteins that regulate the remodeling of the actin cytoskeleton have been proposed [3,4,5,6,7]. Among the various ROCKs, ROCK1 (ROKβ) and ROCK2 (ROKα) share homologous amino acid compositions at the carboxyl termini, the catalytic kinase domain, and the Rho-binding domain in addition to a distinct coiled-coil region [8,9]. ROCK1 and ROCK2 function to regulate the organization of the actin cytoskeleton, differentiation, apoptosis, glucose metabolism, cell adhesion/motility, and inflammation [10,11,12]. ROCKs are also expressed in ocular and peri-ocular tissues, including the trabecular meshwork (TM), ciliary muscles, and the retina [8,9], and also play significant roles in several ocular diseases including cataracts, retinopathy, corneal dysfunction [3,4,13,14,15,16], and glaucoma [17,18]. In our previous study, to study the effects of ROCK-is toward human TM, we developed three-dimensional (3D) drop cell cultures using transforming growth factor-β2 (TGF-β2)-treated human TM (HTM) cells [19], as an in vivo model of primary open glaucoma [20,21,22]. The results demonstrated that pan-ROCK-is, Rip and Y27632 exert significant suppressive effects on the TGF-β2-induced increase in fibrosis [19]. Alternatively, an increase in the stiffness of TM cells caused by the overexpression of ECM such as α-SMA was also reported in human patients treated with dexamethasone (DEX) as well as DEX-treated human TM cell cultures [23]. This finding prompted us to examine the inhibitory effects of ROCK1 and ROCK2 inhibition toward DEX-treated human TM cells using our 3D spheroid cultures.

Therefore, in the current study, to elucidate the role of ROCK1 and ROCK2 toward steroid-induced glaucomatous TM, the effects of the pan-ROCK-i, Rip, and a selective ROCK2 inhibitor (ROCK2-i), KD025 on DEX-stimulated 2D- and 3D-cultured HTM cells were studied. The investigation involved the following issues: transendothelial electron resistance (TEER) measurements and the fluorescein isothiocyanate (FITC)–dextran permeability of 2D-cultured HTM monolayers, the physical properties of the 3D spheroid, including size and stiffness, and the expression of major extracellular matrix (ECM) molecules, namely, collagen (COL) 1, 4 and 6, fibronectin (FN) and α-smooth muscle actin (αSMA), and their modulators, tissue inhibitors of matrix proteinase (TIMP) 1–4, and matrix metalloproteinases (MMP) 2, 9 and 14 (2D and 3D).

## 2. Results

### 2.1. Effects of Pan-ROCK-i, Rip and ROCK2-i, KD025 on TEER and FITC–Dextran Permeability Values for the 2D DEX-Treated HTM Monolayers

In the presence of pan ROCK-i, ripasudil (Rip) or ROCK2-i, KD025 toward DEX-treated HTM cells, the barrier function by transendothelial electron resistance (TEER) measurements and the fluorescein isothiocyanate (FITC)–dextran permeability of 2D-cultured DEX-treated HTM cell monolayers was evaluated. As shown in Figure 1, the TEER values and FITC–dextran permeability were substantially increased and decreased, respectively, upon exposure to 250 nM DEX, and the DEX-induced effects were significantly suppressed by Rip, but not by KD025. This result indicates that DEX induced an effect in the 2D-cultured HTM monolayers, as evidenced by an increase in the TEER value and a decrease in the FITC–dextran permeability, and that these values were exclusively inhibited by ROCK1 inhibition.

### 2.2. Effects of Pan-ROCK-i, Rip and ROCK2-i, KD025 on the Physical Properties, Size and Stiffness, of the 3D DEX-Treated HTM Spheroids

As shown in Figure 2, the 3D spheroid sizes of each experimental conditions were substantially and time-dependently reduced during the 6-day 3D HTM culture, and those down-sizing effects were further enhanced by 250 nM DEX at Day 3 and Day 6. In the presence of Rip or KD025, these effects were suppressed from Day 3 to Day 6 or Day 1 to Day 3, respectively (Figure 2). On the contrary, the stiffness of the 3D HTM spheroids was significantly increased upon the administration of DEX, and these values were also substantially affected and differently by Rip or KD025. That is, Rip significantly enhanced the DEX-induced increase in stiffness, whereas KD025 substantially caused a decrease in stiffness (Figure 3). These results suggest that the inhibition of ROCK1 and ROCK2 significantly affected the course of the 3D culture at different times, but also induced different effects toward the physical properties of the DEX-treated 3D HTM spheroids.

### 2.3. Effects of Pan-ROCK-i, Rip and ROCK2-i, KD025 on the Gene Expressions of ECM and ECM Regulatory Factors of the 2D and 3D DEX-Treated HTM Cells

To elucidate the underlying mechanisms responsible for the drug-induced effects of Rip and KD025 toward the above 2D and 3D DEX-treated HTM cells, the mRNA expression of ECM and their regulatory molecules including TIMPs and MMPs, as possible factors affecting the physical properties of the 3D HTM spheroids, were evaluated (Figure 4, Figure 5 and Figure 6). Upon administering 250 nM DEX, *COL6* (3D), *αSMA* (3D), *TIMP1* (3D), *TIMP2* (2D) and *MMP2* (2D and 3D) were significantly downregulated, and *MMP14* (3D), or *FN* (2D and 3D) and *TIMP4* (3D) were significantly upregulated. The addition of Rip caused a decrease in *COL1* (2D) and *TIMP3* (2D), and an increase in *TIMP3* expressions, respectively. In contrast, the addition of KD025 induced the substantial down-regulation of *COL1* (2D and 3D), *COL4* (3D), *COL6* (3D), *FN* (3D), *TIMP1*, *3* and *4* (3D) and *MMP2* (3D) and the substantial upregulation of *FN* (2D), α*SMA* (3D) and *MMP9* (2D and 3D), respectively. These observations indicate that Rip and KD025 affected the gene expressions of ECM and their modulators, MMPs and TIMPs, in different manners and these gene expressions were also different between 2D and 3D cultures of HTM. Nevertheless, no correlation was observed between these fluctuations of the mRNA expression levels and opposite effects of the physical properties, size and stiffness by Rip and KD025. Therefore, this suggested that such opposite effects by these ROCK-is may be caused by several combinations of these possible factors or by additional factors that were not evaluated.

## 3. Discussion

In contrast to conventional 2D cell culture, 3D spheroid cell cultures have recently received more attention for being suitable applications for in vivo diseases models, including steroid-induced glaucoma [24,25]. Vernazza et al. compared the chronic stress exposure-induced cellular responses between the 2D- and 3D-cultured HTM, and concluded that the 3D-cultured HTM cells are much better with respect to sensitivities against intracellular reactive oxidative species production induced by a hydrogen peroxide treatment [26]. Torrejon et al. also established an in vivo steroid-induced glaucomatous TM model using a scaffold-assisted 3D cell culture and found that ECM alterations in this model can be significantly modified by a pan ROCK-inhibitor, Y27632 [27]. Nevertheless, such 3D culture techniques may have some limitations in terms of mimicking the physiological and pathological conditions of human TM due to the presence of unnecessary scaffolds. To avoid such unnecessary scaffolds, we recently developed a 3D cell drop culture method as in vivo models for Graves’ orbitopathy [28], deepening upper eyelid sulcus (DUES) [29,30]. In our earlier pilot study using this method, we found that TGF-β2 significantly induced the downsizing and stiffness of 3D HTM spheroids, and that these effects were substantially inhibited by pan-ROCK-is [19].

Although ROCK1 and ROCK2 were initially thought to be ubiquitously expressed, and therefore both function in similar roles [8,31], siRNA knockdown studies have shown some difference in their functions in fibroblasts, that is, ROCK1 and ROCK2 induce the assembly of fibronectin matrices at the cell surface in different manners during the actin cytoskeleton-mediated assembly of the extracellular matrix [32,33]. In addition, the knockdown of ROCK1 in keratinocytes induced a decrease in the adhesion of cells to fibronectin, although in contrast, those of the ROCK2 stimulated this adhesion [34,35]. Therefore, these collective observations suggest that ROCK1 and ROCK2 may be localized in various tissues and organs where they play different roles. For example, only ROCK1 is cleaved by caspase-3 during apoptosis [36,37] and smooth muscle-specific basic calponin is phosphorylated by ROCK2, but not by ROCK1 [38]. Furthermore, an analysis by expressed sequence tag (EST) using the tissue-specific gene expression and regulation (TiGer) database [39] indicated that the distributions of ROCK1 and ROCK2 are similar but substantially different in some specific organs and/or tissues. For example, ROCK1 is predominantly expressed in the thymus and blood with little to no ROCK2 expression, and ROCK2 is largely expressed within cardiac and brain tissues in addition to the eye [8,40,41]. Thus, these collective findings led us to speculate that the ROCK-is induced hypotensive effects of IOP might be responsible for the inhibition of ROCK2 rather than ROCK1. In the present study, to verify this speculation, the effects of pan-ROCK-i, Rip and a specific inhibitor of ROCK2, KD025 were compared with each other using DEX-treated 2D and 3D HTM cells. However, quite interestingly, Rip and KD025 caused opposite effects. Although, at time of writing, the precise mechanisms responsible for causing such contrasting drug-induced effects remain to be elucidated, a paradoxical effect toward adipogenesis was recognized by KD025 [42]. Therefore, additional research using specific inhibitors or siRNA to discover possible factors related to ROCK1 and ROCK2 signaling will be required to solve this issue.

## 4. Materials and Methods

### 4.1. Human Trabecular Meshwork (HTM) Cells

Immortalized human trabecular meshwork (HTM) cells were purchased from Applied Biological Materials Inc., Richmond, BC, Canada. In advance of the current study described below, the HTM cells that were used were confirmed to be TM cells by the fact that the mRNA expression of myocilin was upregulated in response to dexamethasone according to the consensus recommendations for TM cells described by Keller et al. [43].

### 4.2. 2D and 3D Cultures of Human Trabecular Meshwork (HTM) Cells

The 2D and 3D cultures of HTM cells were prepared as described in a previous report [19]. Briefly, the 2D-cultured HTM cells were further processed to produce 3D spheroid culture using a hanging droplet spheroid (3D) culture plate (# HDP1385, Sigma-Aldrich, St. Louis, MO, USA) 6 days. For the evaluation of the drug efficacy of ROCK-is on DEX-treated 3D HTM spheroids, a mixture of 250 nM DEX and ROCK-i, 10 µM Rip or KD025 were added at Day 1, and half of the medium (14 μL) in each well was exchanged daily.

### 4.3. Transendothelial Electron Resistance (TEER) Measurements and the Fluorescein Isothiocyanate (FITC)–Dextran Permeability of 2D-Cultured HTM Monolayers

TEER measurements of the monolayered 2D-cultured HTM cells were performed as described previously [44] using a 12-well plate for TEER (0.4 μm pore size and 12 mm diameter; Corning Transwell, Sigma-Aldrich, St. Louis, MO, USA). Briefly, after reaching approximately 80% confluence, 250 nM DEX and ROCK-i, and 10 µM Rip or KD025 were added to the apical side of the wells (Day 1), and the sample was cultured until Day 6. The concentrations of DEX and ROCK-is were confirmed to be the optimum concentrations based on the efficiency of myocilin expression and findings reported in our previous study [19], respectively.

On Day 6, the wells were washed twice with PBS, and TEER (Ωcm^2^) measurements using an electrode (KANTO CHEMICAL CO. INC., Tokyo, Japan) [19] or fluorescein isothiocyanate (FITC)–dextran permeability measurements were made. In terms of FITC–dextran permeability, a 50 μmol/L solution of FITC–dextran (Sigma-Aldrich, St. Louis, MO, USA) was added to the well basal compartments of the culture and the culture medium from the apical compartment was collected at 60 min for the above different experimental conditions. The concentrations of the FITC–dextran were measured using a multimode plate reader (Enspire; Perkin Elmer, Waltham, MA, USA) at an excitation wavelength of 490 and an emission wavelength of 530 nm. The fluorescence intensity of the control medium was used as the background concentration.

### 4.4. Physical Property, Size and Solidity Measurements of 3D Spheroids

As described previously, the 3D spheroid configuration was observed by phase-contrast (PC, Nikon ECLIPSE TS2; Tokyo, Japan) and the mean size of each 3D spheroid defined as the largest cross-sectional area (CSA) was determined using Image-J software version 1.51n (National Institutes of Health, Bethesda, MD) [19,28]. The physical stiffness analysis of the 3D HTM spheroids was performed using a micro-squeezer (MicroSquisher, CellScale, Waterloo, ON, Canada) as previously reported [19,28]. Briefly, under several conditions as above, a single spheroid at Day 6 was moved onto a 3 mm square microplate, and a compression plate was placed on the top of the spheroid. The 3D spheroid was then compressed by the downward movement of the compression plate to cause a 50% deformity of the 3D spheroid during 20 s under monitoring by a microscopic camera. The force required to achieve a deformation of 50% was measured through the cantilever, and the data are expressed as force/displacement (μN/μm).

### 4.5. Quantitative PCR

Total RNA extraction, reverse transcription and the following real-time PCR with the Universal Taqman Master mix using a StepOnePlus instrument (Applied Biosystems/Thermo Fisher Scientific, Waltham, MA, USA) were performed as described previously [19]. The respective cDNA values are shown as fold-change relative to the control of normalized housekeeping gene 36B4 (*Rplp0*). Sequences of primers and Taqman probes used are as described in Table 1.

### 4.6. Statistical Analysis

By statistical analyses using Graph Pad Prism 8 (GraphPad Software, San Diego, CA, USA), statistical significance with a confidence level greater than 95% by a two-tailed Student’s t-test or two-way analysis of variance (ANOVA) followed by a Tukey’s multiple comparison test was performed as described previously [19].

## Figures and Tables

**Figure 1 molecules-26-06382-f001:**
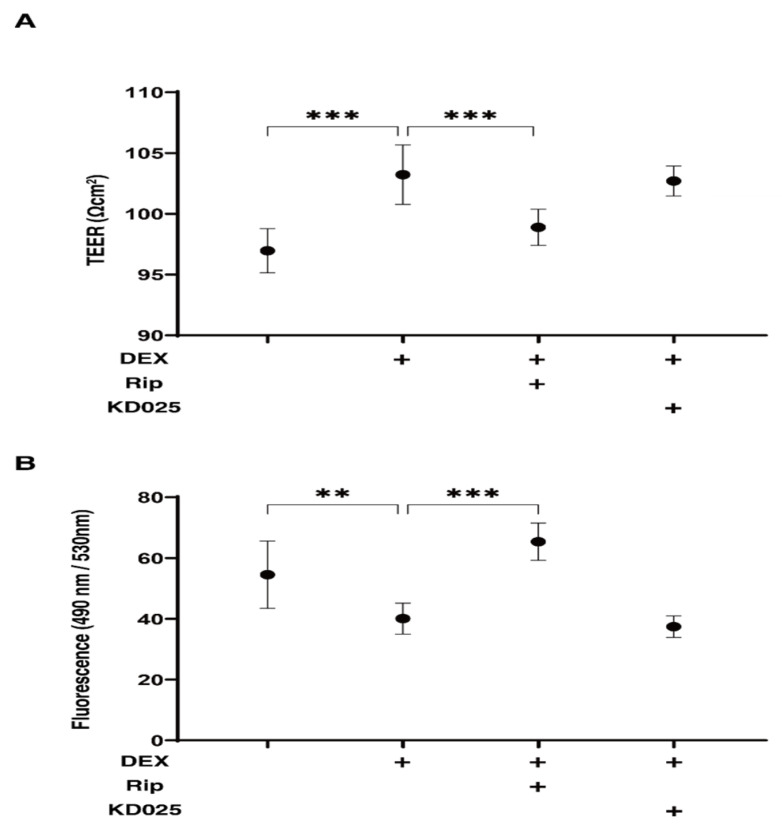
Effects of ROCK-is on transendothelial electrical resistance (TEER) (**A**) and FITC–dextran permeability (**B**) of DEX-treated 2D culture of HTM cell monolayers. In the presence of 10 µM ripasudil (Rip) or KD025, the barrier function (Ωcm^2^) and permeability of DEX (250 nM) untreated or treated 2D-cultured HTM monolayers, TEER (**A**) and FITC–dextran permeability (**B**) were evaluated, respectively. All experiments were performed in triplicate using fresh preparations (*n* = 4). Data are presented as the arithmetic mean ± standard error of the mean (SEM). ** *p* < 0.01, *** *p* < 0.005 (ANOVA followed by a Tukey’s multiple comparison test).

**Figure 2 molecules-26-06382-f002:**
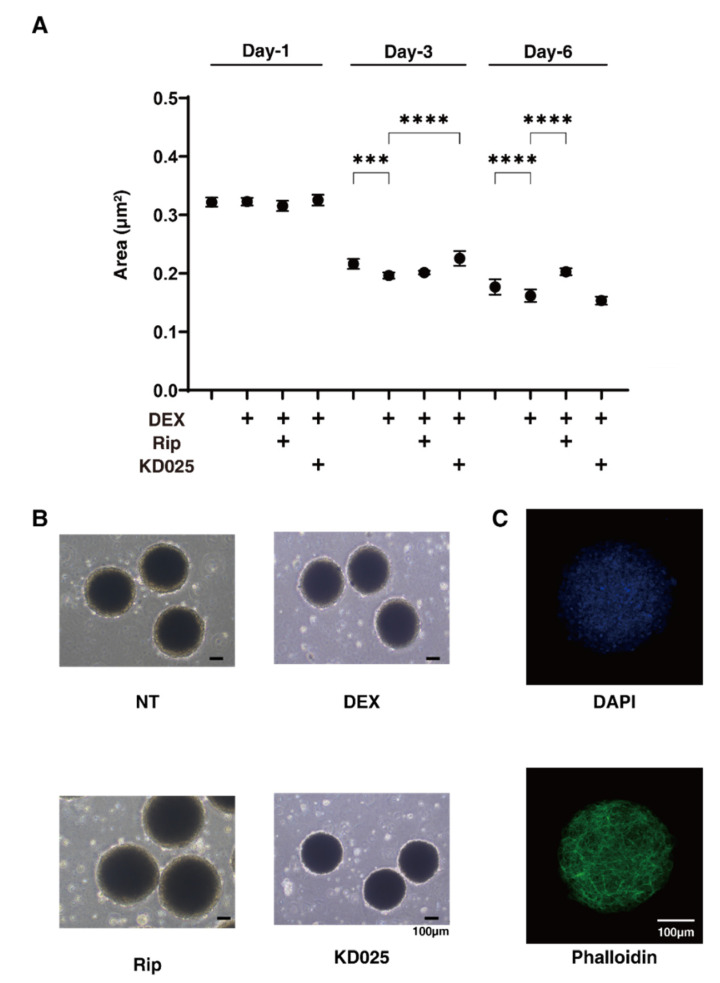
Changes in DEX-treated 3D HTM spheroid size at Days 1, 3, or 6 in the presence and absence of ripasudil or KD025, and a representative non-treated 3D spheroid image stained with DAPI and phalloidin (**C**). At Days 1, 3, or 6, the mean sizes of HTM 3D spheroids (non-treated control; NT) and those treated with 250 nM DEX were plotted in the absence or presence of 10 µM ripasudil (Rip) or KD025 (**A**). (**B**) shows representative phase-contrast microscope images of the 3D HTM spheroids at Day 6 under several conditions. (**C**) shows a representative image immune-stained with DAPI and phalloidin of non-treated 3D HTM spheroid. These experiments were performed in triplicate using fresh preparations (*n* = 10–15). Data are presented as the arithmetic mean ± standard error of the mean (SEM). *** *p* < 0.005, **** *p* < 0.001 (ANOVA followed by a Tukey’s multiple comparison test). Scale bar; 100 µm.

**Figure 3 molecules-26-06382-f003:**
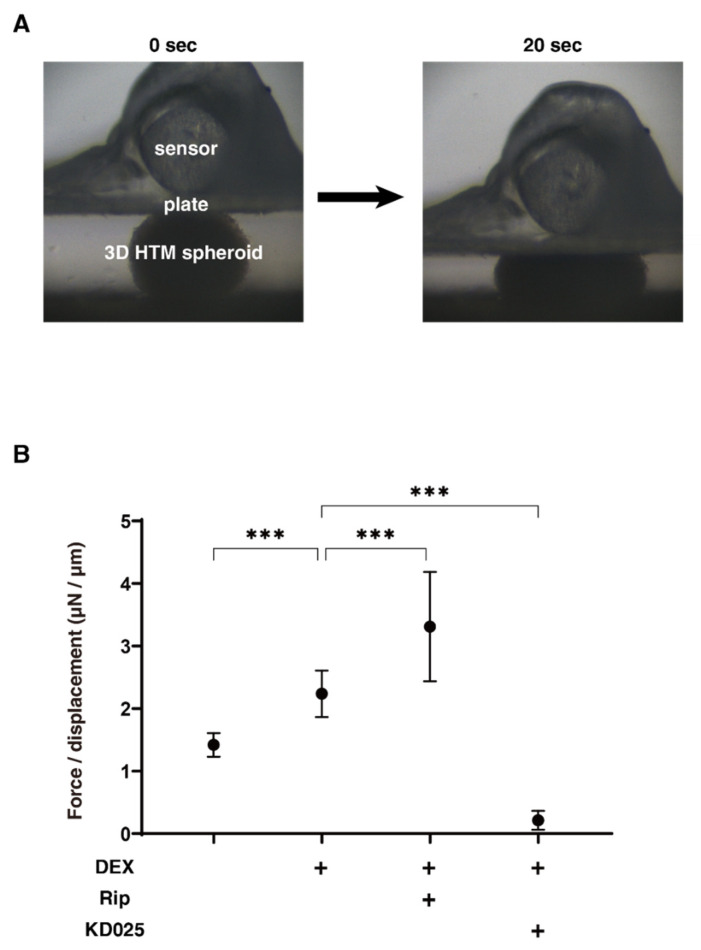
Physical solidity of 3D HTM spheroids. (**A**) Representative photos demonstrating the micro-squeezer analysis measured over a period of 20 s. On Day 6, the force (μN) required to induce a 50% deformity of every 1 out of 15–20 freshly prepared 3D HTM spheroids (non-treated control) and those treated with 250 nM DEX in the absence or presence of 10 µM ROCK-i, ripasudil (Rip) or KD025 were measured, and force/displacement (μN/μm) values were plotted in (**B**). These experiments were performed in triplicate using fresh preparations (*n* = 10). *** *p* < 0.005 (ANOVA followed by a Tukey’s multiple comparison test).

**Figure 4 molecules-26-06382-f004:**
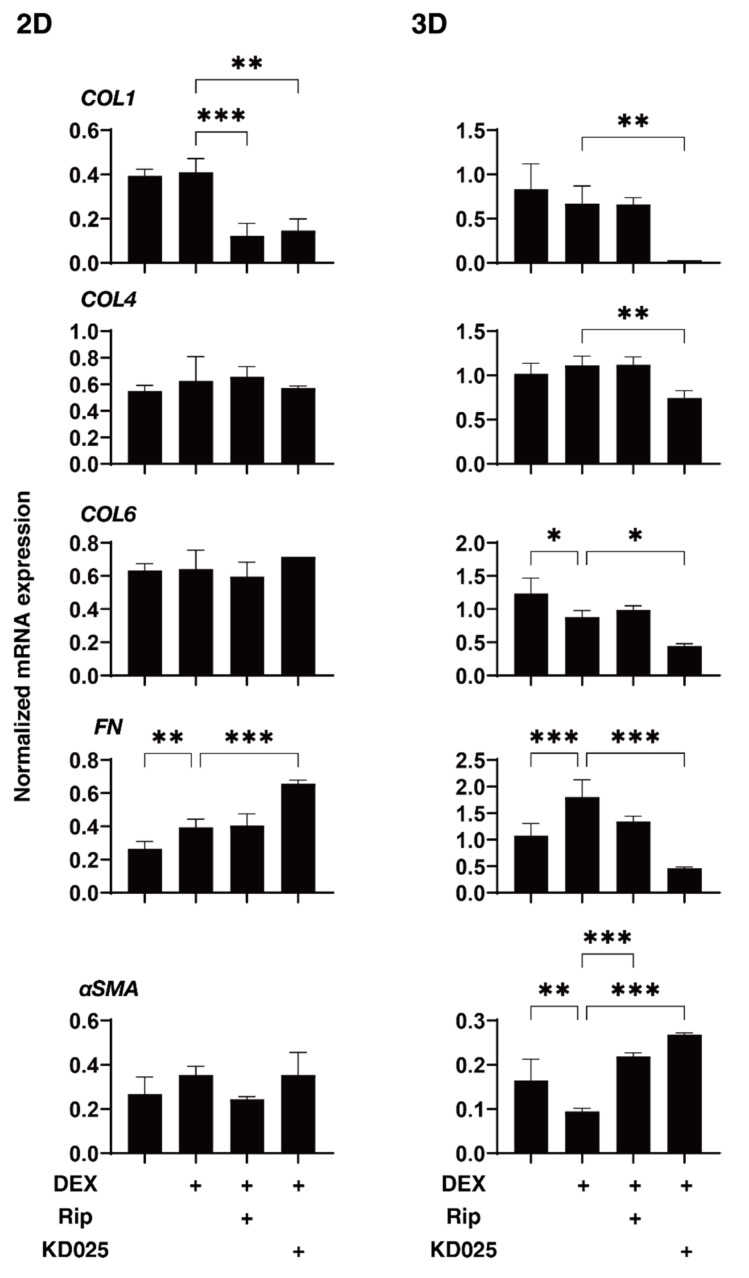
mRNA expression of ECM molecules in 2D- and 3D-cultured HTM cells. On Day 6, HTM 2D cells and 3D spheroids (non-treated control; NT) and those treated with 250 nM DEX in the absence or presence of 10 µM ripasudil (Rip) or KD025 were subjected to qPCR analysis to estimate the expression of mRNA in ECMs (*COL1*, *COL4*, *COL6*, *FN* and *α-SMA*). All experiments were performed in duplicate using 3 different confluent 6-well dishes (2D) or 15–20 freshly prepared 3D HTM spheroids (3D) in each experimental condition. Data are presented as the arithmetic mean ± standard error of the mean (SEM). * *p* < 0.05, ** *p* < 0.01, *** *p* < 0.005 (ANOVA followed by a Tukey’s multiple comparison test).

**Figure 5 molecules-26-06382-f005:**
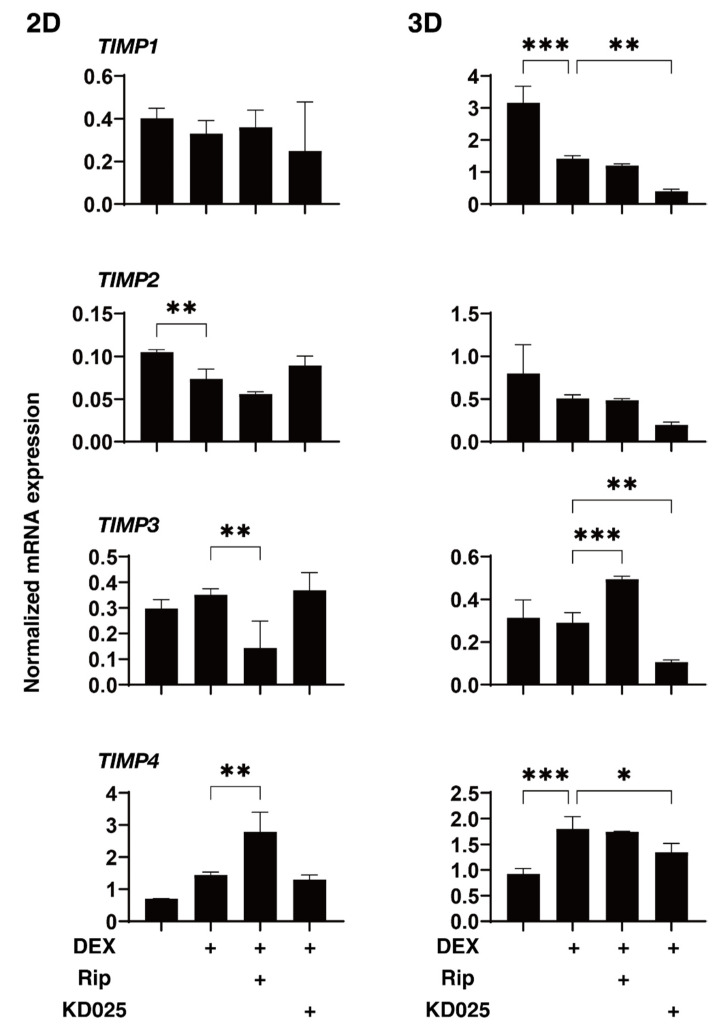
mRNA expression of TIMPs in 2D- and 3D-cultured HTM cells. On Day 6, HTM 2D cells and 3D spheroids (non-treated control; NT) and those treated with 250 nM DEX in the absence or presence of 10 µM ripasudil (Rip) or KD025 were subjected to qPCR analysis to estimate the expression of mRNA in *TIMP1–4*. All experiments were performed in duplicate using 3 different confluent 6-well dishes (2D) or 15–20 freshly prepared 3D HTM spheroids (3D) in each experimental condition. Data are presented as the arithmetic mean ± standard error of the mean (SEM). * *p* < 0.05, ** *p* < 0.01, *** *p* < 0.005 (ANOVA followed by a Tukey’s multiple comparison test).

**Figure 6 molecules-26-06382-f006:**
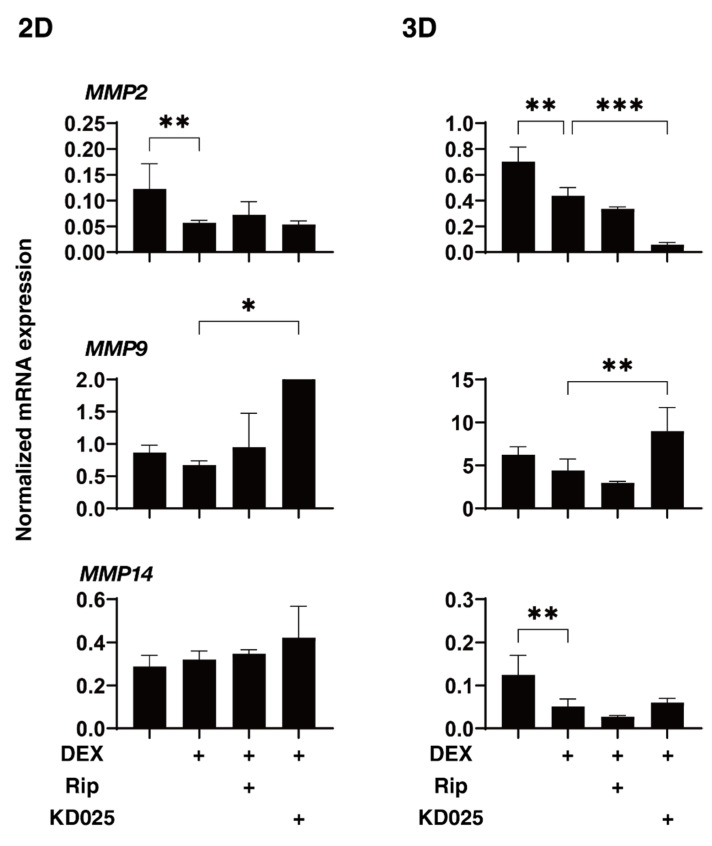
mRNA expression of MMPs in 2D- and 3D-cultured HTM cells. On Day 6, HTM 2D cells and 3D spheroids (non-treated control; NT) and those treated with 250 nM in the absence or presence of 10 µM ripasudil (Rip) or KD025 were subjected to qPCR analysis to estimate the expression of mRNA of *MMP2*, *9* and *14*. All experiments were performed in duplicate using 3 different confluent 6-well dishes (2D) or 15–20 freshly prepared 3D HTM spheroids (3D) in each experimental condition. Data are presented as the arithmetic mean ± standard error of the mean (SEM). * *p* < 0.05, ** *p* < 0.01, *** *p* < 0.005 (ANOVA followed by a Tukey’s multiple comparison test).

**Table 1 molecules-26-06382-t001:** Sequences of primers used in the qPCR.

Sequence			Exon Location	RefSeq Number
human RPLP0	Probe Primer2 Primer1	5′-/56-FAM/CCCTGTCTT/ZEN/CCCTGGGCATCAC/3IABkFQ/-3′ 5′-TCGTCTTTAAACCCTGCGTG-3′ 5′-TGTCTGCTCCCACAATGAAAC-3′	2–3	NM_001002
human COL1A1	Probe Primer2 Primer1	5′-/56-FAM/TCGAGGGCC/ZEN/AAGACGAAGACATC/3IABkFQ/-3′ 5′-GACATGTTCAGCTTTGTGGAC-3′ 5′-TTCTGTACGCAGGTGATTGG-3′	1–2	NM_000088
human COL4A1	Probe Primer2 Primer1	5′-/56-FAM/TCATACAGA/ZEN/CTTGGCAGCGGCT/3IABkFQ/-3′ 5′-AGAGAGGAGCGAGATGTTCA-3′ 5′-TGAGTCAGGCTTCATTATGTTCT-3′	51–52	NM_001845
human COL6A1	Primer2 Primer1	5′-CCTCGTGGACAAAGTCAAGT-3′ 5′-GTGAGGCCTTGGATGATCTC-3′	2–3	NM_001848
human FN1	Primer2 Primer1	5′-CGTCCTAAAGACTCCATGATCTG-3′ 5′-ACCAATCTTGTAGGACTGACC-3′	3–4	NM_212482
human αSMA	Probe Primer2 Primer1	5′-/56-FAM/AGACCCTGT/ZEN/TCCAGCCATCCTTC/3IABkFQ/-3′ 5′-AGAGTTACGAGTTGCCTGATG-3′ 5′-CTGTTGTAGGTGGTTTCATGGA-3′	8–9	NM_001613
human TIMP1	Probe Primer2 Primer1	5′-/56-FAM/TCAACCAGA/ZEN/CCACCTTATACCAGCG/3IABkFQ/-3′ 5′-CCTTCTGCAATTCCGACCT-3′ 5′-GCTTGGAACCCTTTATACATCTTG-3′	2–4	NM_003254
human TIMP2	Probe Primer2 Primer1	5′-/56-FAM/TCTCATTGC/ZEN/AGGAAAGGCCGAGG/3IABkFQ/-3′ 5′-GACGTTGGAGGAAAGAAGGA-3′ 5′-TGTGGTTCAGGCTCTTCTTC-3′	3–4	NM_003255
human TIMP3	Probe Primer2 Primer1	5′-/56-FAM/CCTCCTTTA/ZEN/CCAGCTTCTTCCCCAC/3IABkFQ/-3′ 5′-CCTTCTGCAACTCCGACATC-3′ 5′-CGGTACATCTTCATCTGCTTGA-3′	1–3	NM_000362
human TIMP4	Probe Primer2 Primer1	5′-/56-FAM/ACTGAGGAC/ZEN/CTGACCAGTCAAGAGA/3IABkFQ/-3′ 5′-GGTTTGAGAAAGTCAAGGATGTTC-3′ 5′-GTTGCACAGATGGATGAAGAC-3′	3–4	NM_003256
human MMP2	Primer2 Primer1	5′-TCCACCACCTACAACTTTGAG-3′ 5′-GTGCAGCTGTCATAGGATGT-3′	6–7	NM_004530
human MMP9	Primer2 Primer1	5′-ACATCGTCATCCAGTTTGGTG-3′ 5′-CGTCGAAATGGGCGTCT-3′	3–4	NM_004994
human MMP14	Primer2 Primer1	5′-TTCGCCGACTAAGCAGAAG-3′ 5′-CTTGAATTCCTAGACCGCTGT-3′	1–1	NM_004995

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
