# Peer review of "Pan-ROCK and ROCK2 Inhibitors Affect Dexamethasone-Treated 2D- and 3D-Cultured Human Trabecular Meshwork (HTM) Cells in Opposite Manners"

_molecules, 2021, doi:10.3390/molecules26216382_

Round 1

Reviewer 1 Report

Review of the article of Megumi Watanabe, Yosuke Ida, Masato Furuhashi, Yuri Tsugeno, Hiroshi Ohguro  and Fumihito Hikage “Pan-ROCK and ROCK2 inhibitors affect dexamethasone treated 2D and 3D cultured human trabecular meshwork 3 (HTM) cells in opposite manners”. The authors of the article have done a lot of experimental work comparing the effect of ripasudil hydrochloride hydrate, a pan-ROCK inhibitor, and KD025, a selective inhibitor of ROCK2 protein kinases, on human trabecular meshwork cells treated with dexamethasone. However, the results obtained, in my opinion, are not well represented.

Comments

      In the abstract, the authors list the multiple effects of pan-ROCK and ROCK-2 kinase inhibitors on dexamethasone-treated human trabecular reticulum cells using a huge number of contractions. It is very difficult to read and understand. To facilitate understanding of the content, authors should indicate the purpose of the work and its meaning. For example, the authors may very briefly state that they have used dexamethasone-treated trabecular meshwork cells as a model of glaucoma and ROCK inhibitors as a new type of ocular antihypertensive drug used in the treatment of glaucoma. At the conclusion the authors write: “The present findings indicate that 2D and 3D cell cultures can be useful for evaluating for drug induced effects of Rip and KD025 with respect to DEX treated HTM cells”. Is this the main conclusion of the work?!!!

      The introduction is also replete with abbreviations, fact sheets, and long phrases.

Could anyone understand this sentence: “Since it was demonstrated that transforming growth factor-β2 (TGF-β2) stimulates a significant increase in the expression of ECM molecules by TM cells, and induces an irreversible cross-linking of TM fibronectin, we used our developed 3D culture technique and confirmed that it exerts significant suppressive effects on the increase in fibrosis in TGF-β2 treated 3D HTM spheroids caused by pan-ROCK-is, Rip and Y2763219”?  (Lines 45- 49)

Line 51: “the over expression of ECM such as α-SMA was also reported” - α-smooth muscle actin (SMA) is not an ECM molecule.

Figure 1 A, figure 2 (upper panel) and figure 3 B.

Why authors compare Dex+KD025 with Dex + Rip?

I think they have compare Dex+KD025 with Dex.

Figure 2

The authors mentioned panel A, B, C in the figure capture but not in the figure itself.

Lines 87-88: “Fig. 2 indicated that their sizes (spheroid sizes) were substantially reduced during the 3D culture, and such down-sizing effects……..”. The size of spheroids in a 3D culture is reduced compared to what? Figure 2 and text does not contain such information.

In paragraphs 86-95, the text is duplicated.

Figure 3 A: “sphenoid”

Lines 119-121: again, α-smooth muscle actin (SMA) is cytosolic protein but not an extracellular matrix protein (ECM).  

Figure 5, 6, 7

The time of exposure to dexamethasone on cells is not indicated in the text, nor in the methods, or in the figure captions.

Lines 118-129

This paragraph presents a listing of the data without attempting to generalize the results and indicate their physiological significance. Only in the end of discussion (lines 195-196) the authors wrote that the diversity between the Rip and KD025-induced effects may be caused by different alterations in the gene expression of ECM molecules and their modulators, TIMPs by Rip and MMPs by KD025. In order to understand the difference between the tested ROCK inhibitors, it would be useful to decipher the abbreviations for metalloproteinases and tissue metalloproteinase inhibitors and discuss their role in cell physiology. It can help very much to understand the results of the article.

      During the discussion, the authors list the results again instead of discussing their implications in cell physiology and biochemistry or pharmacology. Non-specialist readers would be interested to know briefly how dexamethasone alters cell physiology and why dexamethasone-treated cells are used as cell models in glaucoma. Authors should discuss the difference in their results for 2D and 3D cultures. The article should be rewritten, while significantly reducing the number of abbreviations, avoiding long phrases and long lists of facts without explaining their meaning. Authors must draw conclusions in which to indicate the significance of the results obtained.

Author Response

Dear Editor,

Thank you very much for the constructive comments concerning our manuscript; " Pan-ROCK and ROCK2 inhibitors affect dexamethasone treat-ed 2 D and 3D cultured human trabecular meshwork (HTM) cells in opposite manners “. We carefully examined all of the comments from the Reviewer and have made a series of specific changes to our manuscript as follows;

Reviewer 1

Review of the article of Megumi Watanabe, Yosuke Ida, Masato Furuhashi, Yuri Tsugeno, Hiroshi Ohguro and Fumihito Hikage “Pan-ROCK and ROCK2 inhibitors affect dexamethasone treated 2D and 3D cultured human trabecular meshwork 3 (HTM) cells in opposite manners”. The authors of the article have done a lot of experimental work comparing the effect of ripasudil hydrochloride hydrate, a pan-ROCK inhibitor, and KD025, a selective inhibitor of ROCK2 protein kinases, on human trabecular meshwork cells treated with dexamethasone. However, the results obtained, in my opinion, are not well represented.

  1. In the abstract, the authors list the multiple effects of pan-ROCK and ROCK-2 kinase inhibitors on dexamethasone-treated human trabecular reticulum cells using a huge number of contractions. It is very difficult to read and understand. To facilitate understanding of the content, authors should indicate the purpose of the work and its meaning. For example, the authors may very briefly state that they have used dexamethasone-treated trabecular meshwork cells as a model of glaucoma and ROCK inhibitors as a new type of ocular antihypertensive drug used in the treatment of glaucoma. At the conclusion the authors write: “The present findings indicate that 2D and 3D cell cultures can be useful for evaluating for drug induced effects of Rip and KD025 with respect to DEX treated HTM cells”. Is this the main conclusion of the work?!!!

Answer; As suggested, abstract was changed more simpler; “Effects of a pan-ROCK-inhibitor, ripasudil (Rip), and a ROCK2 inhibitor, KD025 on dexame-thasone (DEX)-treated human trabecular meshwork (HTM) cells as a model of steroid induced glaucoma were investigated. In the presence of Rip or KD025, DEX-treated HTM cells were sub-jected to permeability analysis of 2D monolayer by transendothelial electrical resistance (TEER) and FITC-dextran permeability, physical properties, size and stiffness analysis (3D), and qPCR of extracellular matrix (ECM) and their modulators. DEX resulted in a significant increase of the permeability, as well as a large and stiffed 3D spheroid, and those effects were inhibited by Rip. In contrast, KD025 exerted opposite effects on the physical properties (down-sizing and soften-ing). Furthermore, DEX induced several changes of gene expressions of ECM and their modulators were also modulated differently by Rip and KD025. The present findings indicate that Rip and KD025 induced opposites effects toward 2D and 3D cell cultures of DEX treated HTM cells.”.

  1. The introduction is also replete with abbreviations, fact sheets, and long phrases. Could anyone understand this sentence: “Since it was demonstrated that transforming growth factor-β2 (TGF-β2) stimulates a significant increase in the expression of ECM molecules by TM cells, and induces an irreversible cross-linking of TM fibronectin, we used our developed 3D culture technique and confirmed that it exerts significant suppressive effects on the increase in fibrosis in TGF-β2 treated 3D HTM spheroids caused by pan-ROCK-is, Rip and Y2763219”? (Lines 45- 49)

Answer; As suggested, the introduction was changed more simpler; “1. Introduction Quite recently, a new type of ocular hypotensive drug, ripasudil hydrochloride hydrate (Rip), a Rho-associated coiled-coil containing protein kinase (ROCKs) inhibitor (ROCK-i), has been made available for use in our clinic as an additional option for an anti-glaucoma medication1,2. As a possible mechanism for decreasing IOP by inhibiting ROCKs, ubiquitous downstream effector proteins that regulate the remodeling of the actin cytoskeleton have been proposed3-7. Among the various ROCKs, ROCK1 (ROKβ) and ROCK2 (ROKα share homologous amino acid compositions at the carboxyl termini, the catalytic kinase domain and the Rho-binding domain in addition to a distinct coiled-coil region8,9. Functionally, ROCK1 and ROCK2 function to regulate the organization of the ac-tin cytoskeleton, differentiation, apoptosis, glucose metabolism, cell adhesion/motility, and inflammation10-12. ROCKs are also expressed in ocular and peri-ocular tissues, including the trabecular meshwork (TM), ciliary muscles, and the retina8,9, and also play significant roles in several ocular diseases including cataracts, retinopathy, and corneal dysfunction3,4,13-16 in addition to glaucoma17,18. In our previous study, to study effects of ROCK-is toward human TM, we developed three-dimension (3D) drop cell cultures using transforming growth factor-β2 (TGF-β2) treated human TM (HTM) cells19, as an in vivo model of primary open glaucoma20-22. The results demonstrated that pan-ROCK-is, Rip and Y27632 exert significant suppressive effects on the TGF-β2 induced increase in fibro-sis19. Alternatively, an increase in the stiffness of TM cells caused by the over expression of ECM such as α-SMA was also reported in human patients treated dexamethasone (DEX) as well as DEX-treated human TM cell cultures23. This finding prompted us to examine the inhibitory effects of ROCK1 and ROCK2 inhibition toward DEX treated human TM cells using our 3D spheroid cultures.

Therefore, in the current study, to elucidate the role of ROCK1 and ROCK2 toward steroid induced glaucomatous TM, the effects of the pan-ROCK-i, Rip and a selective ROCK2 inhibitor (ROCK2-i), KD025 on DEX stimulated 2D and 3D cultured HTM cells were studied. The investigation involved the following issues; transendothelial electron resistance (TEER) measurements and the fluorescein isothiocyanate (FITC)-dextran per-meability of 2D cultured HTM monolayers, the physical properties of the 3D spheroid, including size and stiffness, and the expression of major extracellular matrix (ECM) molecules, namely, collagen (COL) 1, 4 and 6, fibronectin (FN) and α-smooth muscle actin (αSMA), and their modulators, tissue inhibitors of matrix proteinase (TIMP) 1-4, and matrix metalloproteinases (MMP) 2, 9 and 14 (2D and 3D).”.

  1. Line 51: “the over expression of ECM such as α-SMA was also reported” - α-smooth muscle actin (SMA) is not an ECM molecule.

Answer; Thank you for this comment. As pointed out, these were changed to” the over expression of α-SMA was also reported”.

  1. Figure 1 A, figure 2 (upper panel) and figure 3 B. Why authors compare Dex+KD025 with Dex + Rip? I think they have compare Dex+KD025 with Dex.

Answer; Basically, we performed statistical analysis of all possible pairs. As pointed out, comparison between Dex+KD025 with Dex + Rip was deleted.

  1. Figure 2. The authors mentioned panel A, B, C in the figure capture but not in the figure itself. Lines 87-88: “Fig. 2 indicated that their sizes (spheroid sizes) were substantially reduced during the 3D culture, and such down-sizing effects……..”. The size of spheroids in a 3D culture is reduced compared to what? Figure 2 and text does not contain such information.

Answer; As suggested, panel A, B, C were also included within the figure in addition to the legends. In addition, missing panel C was included in Fig. 2. In terms of the reducing spheroid size, to avoid ambiguity, the corresponding phrase was changed; “As shown in Fig. 2, the 3D spheroid sizes of each experimental conditions were sub-stantially and time-dependently reduced during the 6-day 3D HTM culture”.

  1. In paragraphs 86-95, the text is duplicated.

Answer; As suggested, this paragraph was changed to avoid duplicates; “As shown in Fig. 2, the 3D spheroid sizes of each experimental conditions were sub-stantially and time-dependently reduced during the 6-day 3D HTM culture, and those down-sizing effects were further enhanced by 250 nM DEX at Day 3 and Day 6. In the presence of Rip or KD025, these effects were suppressed during Day 3 to Day 6 or Day 1 to Day 3, respectively (Fig 2). To the contrary, the stiffness of the 3D HTM spheroids was significantly increased upon the administration of DEX, and these values were also substantially affected and differently by Rip or KD025. That is, Rip significantly enhanced the DEX-induced increase in stiffness, while KD025 substantially caused a decrease in stiff-ness (Fig. 3). These results suggest that the inhibition of ROCK1 and ROCK2 significantly affected the course of the 3D culture at different times, but also induced different effects toward the physical properties of the DEX-treated 3D HTM spheroids.”.

  1. Figure 3 A: “sphenoid”

Answer; As pointed out, that was changed to “spheroid”.

  1. Lines 119-121: again, α-smooth muscle actin (SMA) is cytosolic protein but not an extracellular matrix protein (ECM).

Answer; As pointed out, these phrases were changed to “2.3. Effects of pan-ROCK-i, Rip and ROCK2-i, KD025 on the gene expressions of ECM and their related factors of the 2D and 3D DEX-treated HTM cells.

To elucidate the underlying mechanisms responsible for the drug induced effects of Rip and KD025 toward the above 2D and 3D DEX treated HTM cells, the mRNA expres-sion of ECM molecules and their related factors including aSMA, TIMPs and MMPs were evaluated (Figs. 4-6). Upon administering 250 nM DEX, COL6 (3D), aSMA (3D), TIMP1 (3D), TIMP2 (2D) MMP2 (2D and 3D) were significantly down-regulated, and MMP14 (3D), or FN (2D and 3D) and TIMP4 (3D) were significantly up-regulated.”.

  1. Figure 5, 6, 7. The time of exposure to dexamethasone on cells is not indicated in the text, nor in the methods, or in the figure captions.

Answer; In terms of the exposure to dexamethasone on cells, this information is included within the corresponding method; “4.2. 2D and 3D cultures of human trabecular meshwork (HTM) Cells

2D and 3D cultures of HTM cells were prepared as described in a previous report19. Briefly, the 2D cultured HTM cells were further processed to produce 3D spheroid culture using a hanging droplet spheroid (3D) culture plate (# HDP1385, Sigma-Aldrich) during 6 days. For the evaluation of the drug efficacy of ROCK-is on DEX treated 3D HTM spheroids, a mixture of 250 nM DEX and ROCK-i, 10 µM Rip or KD025 were added at day 1 through day 6, and half of the medium (14 μL) in each well was exchanged daily.”.

  1. Lines 118-129. This paragraph presents a listing of the data without attempting to generalize the results and indicate their physiological significance. Only in the end of discussion (lines 195-196) the authors wrote that the diversity between the Rip and KD025-induced effects may be caused by different alterations in the gene expression of ECM molecules and their modulators, TIMPs by Rip and MMPs by KD025. In rder to understand the difference between the tested ROCK inhibitors, it would be useful to decipher the abbreviations for metalloproteinases and tissue metalloproteinase inhibitors and discuss their role in cell physiology. It can help very much to understand the results of the article.

Answer; As suggested, in addition to a listing of the data, possible interpretation to generalize the corresponding results was included; “To elucidate the underlying mechanisms responsible for the drug induced effects of Rip and KD025 toward the above 2D and 3D DEX treated HTM cells, the mRNA expression of ECM and their regulatory molecules including TIMPs and MMPs, as possible fac-tors affecting physical properties of the 3D HTM spheroids, were evaluated (Figs. 4-6). Upon administering 250 nM DEX, COL6 (3D), aSMA (3D), TIMP1 (3D), TIMP2 (2D) MMP2 (2D and 3D) were significantly down-regulated, and MMP14 (3D), or FN (2D and 3D) and TIMP4 (3D) were significantly up-regulated. The addition of Rip caused a decrease of COL1 (2D) and TIMP3 (2D), and an increase in TIMP3 expressions, respectively. While in contrast, the addition of KD025 induced the substantial down-regulation of COL1 (2D and 3D), COL4 (3D), COL6 (3D), FN (3D), TIMP1, 3 and 4 (3D) and MMP2 (3D) and the substantial up-regulation of FN (2D), αSMA (3D) and MMP9 (2D and 3D), respectively. These observations indicate that Rip and KD025 affected the gene expressions of ECM and their modulators, MMPs and TIMPs, in different manners and these gene expressions were also different between 2D and 3D cultures of HTM. Nevertheless, no correlation was observed between these fluctuations of the mRNA expression levels and opposite effects of the physical properties, size and stiffness by Rip and KD025. Therefore, this suggested that such opposite effects by these ROCK-is may be several combinations of these possible fac-tors or additional factors that were not evaluated, other than one of factors tested.”.

  1. During the discussion, the authors list the results again instead of discussing their implications in cell physiology and biochemistry or pharmacology. Non-specialist readers would be interested to know briefly how dexamethasone alters cell physiology and why dexamethasone-treated cells are used as cell models in glaucoma. Authors should discuss the difference in their results for 2D and 3D cultures. The article should be rewritten, while significantly reducing the number of abbreviations, avoiding long phrases and long lists of facts without explaining their meaning. Authors must draw conclusions in which to indicate the significance of the results obtained.

Answer; As suggested, to change discussion more simpler, 2nd paragraph of discussion was changed; “Although ROCK1 and ROCK2 were initially thought to be ubiquitously expressed, and therefore both function in similar roles31,32, siRNA knockdown studies have shown some difference in their functions in fibroblasts, that is, ROCK1 and ROCK2 induce the assembly of fibronectin matrices at the cell surface in different manners during the actin cytoskeleton mediated assembly of the extracellular matrix33,34. In addition, the knock-down of ROCK1 in keratinocytes induced a decrease in the adhesion of cells to fibronectin, while in contrast, those of the ROCK2 stimulated this adhesion35,36. Therefore, these collective observations suggest that ROCK1 and ROCK2 may be localized in various tissues and organs where they play different roles. For example, only ROCK1 is cleaved by caspase-3 during apoptosis37,38 and smooth muscle-specific basic calponin is phosphorylated by ROCK2, but not by ROCK139. Furthermore, an analysis by expressed sequence tag (EST) using the tissue-specific gene expression and regulation (TiGer) database40 indicated that the distributions of ROCK1 and ROCK2 are similar but, substantially different in some specific specific organs and/or tissues. For example, ROCK1 is predominantly expressed in the thymus and blood with little to no ROCK2 expression, and ROCK2 is largely expressed within cardiac and brain tissues in addition to the eye32,41,42. Thus, these collective findings led us to speculate that the ROCK-is induced hypotensive effects of IOP might be responsible for the inhibition of ROCK2 rather than ROCK1. In the present study, to verify this speculation, the effects of pan-ROCK-i, Rip and a specific inhibitor of ROCK2, KD025 were compared with each other using DEX treated 2D and 3D HTM cells. However, quite interestingly, Rip and KD025 caused opposite effects. Although, as of writing this, the pre-cise mechanisms responsible for causing such drugs induced opposite effects remain to be elucidated, paradoxical effect toward adipogenesis was recognized by KD025. Therefore, additional study using specific inhibitors or siRNA toward possible factors related to ROCK1 and 2 signaling will be required to solve this issue”.

Reviewer 2 Report

In previous studies, the authors demonstrated that TGF-B2 treatment increases the expression of exacellular matrix molecules and induces fibronectin crosslinking in the trabecular meshwork. Using a three-dimensional cell culture technique, they were able to demonstrate that the increase in fibrosis caused by TGF could be reversed by treatment with ROCK's inhibitors. In addition, it is observed that in patients treated with Dexamethasone there is an increase in the stiffness of the trabecular meshwork due to the overexpression of extracellular matrix molecules. The objective of this work was to characterize the role of ROCK1 and 2 in glaucomatous trabecular meshwork induced by steroid.

The authors studied the effect of several ROCK's inhibitors on 2D and 3D cultures of trabecular meshwork cells stimulated with dexamethasone. To do this, they used TEER, FITC-dextran permeability, and RT-RCP techniques.

The results obtained by the authors show that treatment with Dexa increases TEER values and decrease the permeability of FITC-dextran, the effect of dexa is counteracted by co-incubating with Rip but not by KD025. On the other hand, Dexa decrease the size and increase stiffness of the 3D spheroids. Treatment with Rip increases dexa-induced stiffness, while KD025 decreases it.

The dexa-induced expression of genes related to extracellular matrix elements is regulated differently by Rip and by KD025.

In conclusion, ROCK 1 and ROCK 2 modulate differently the TEER values and permeability of 2D cultures and the size and stiffness of 3D spheroids.

The work is well organized and the results are clearly and concisely.

Although the mechanism through which these drugs exert their effects remains to be elucidated, the work provides new results that contribute to building knowledge about the mechanism of action of those inbihitors.

My recommendation is in favor of its publication.

Author Response

Dear Editor,

Thank you very much for the constructive comments concerning our manuscript; " Pan-ROCK and ROCK2 inhibitors affect dexamethasone treat-ed 2 D and 3D cultured human trabecular meshwork (HTM) cells in opposite manners “. We carefully examined all of the comments from the Reviewer and have made a series of specific changes to our manuscript as follows;

In previous studies, the authors demonstrated that TGF-B2 treatment increases the expression of exacellular matrix molecules and induces fibronectin crosslinking in the trabecular meshwork. Using a three-dimensional cell culture technique, they were able to demonstrate that the increase in fibrosis caused by TGF could be reversed by treatment with ROCK's inhibitors. In addition, it is observed that in patients treated with Dexamethasone there is an increase in the stiffness of the trabecular meshwork due to the overexpression of extracellular matrix molecules. The objective of this work was to characterize the role of ROCK1 and 2 in glaucomatous trabecular meshwork induced by steroid.

The authors studied the effect of several ROCK's inhibitors on 2D and 3D cultures of trabecular meshwork cells stimulated with dexamethasone. To do this, they used TEER, FITC-dextran permeability, and RT-RCP techniques.

The results obtained by the authors show that treatment with Dexa increases TEER values and decrease the permeability of FITC-dextran, the effect of dexa is counteracted by co-incubating with Rip but not by KD025. On the other hand, Dexa decrease the size and increase stiffness of the 3D spheroids. Treatment with Rip increases dexa-induced stiffness, while KD025 decreases it.

The dexa-induced expression of genes related to extracellular matrix elements is regulated differently by Rip and by KD025.

In conclusion, ROCK 1 and ROCK 2 modulate differently the TEER values and permeability of 2D cultures and the size and stiffness of 3D spheroids.

The work is well organized and the results are clearly and concisely.

Although the mechanism through which these drugs exert their effects remains to be elucidated, the work provides new results that contribute to building knowledge about the mechanism of action of those inbihitors.

My recommendation is in favor of its publication.

Answer; Thank you so much for such encouraging comments.

Reviewer 3 Report

This study only contains gene expression analysis. Protein expression analysis by western blot is a must to come to a strong conclusion. I reject this at this form. 

Author Response

Dear Editor,

Thank you very much for the constructive comments concerning our manuscript; " Pan-ROCK and ROCK2 inhibitors affect dexamethasone treat-ed 2 D and 3D cultured human trabecular meshwork (HTM) cells in opposite manners “. We carefully examined all of the comments from the Reviewer and have made a series of specific changes to our manuscript as follows;

Reviewer 3

This study only contains gene expression analysis. Protein expression analysis by western blot is a must to come to a strong conclusion. I reject this at this form.

Answer; Thank you for this comment. In terms of protein expression levels but not gene expression levels, I agree with that it will be better to add western blot analysis in addition to qPCR analysis. However, only a week point of our current 3D HTM spheroid culture is that the 3D spheroid contains insufficient amounts of proteins amounts for western blot analysis. Therefore, to perform western blot analysis, at least 50 to 100 spheroids will be required for one analysis. In case to perform western blot analysis in duplicate or triplicates, the results may be mean values of contents within several hundred 3D spheroids. Actually, instead of the western blot analysis, we already performed immunocytochemistry using specific antibodies toward ECMs, and measured the immunostaining densities of every single out of at least 20-30 3D HTM spheroids (ROCK inhibitors beneficially alter the spatial configuration of TGFβ2-treated 3D organoids from a human trabecular meshwork (HTM). Ota C, Ida Y, Ohguro H, Hikage F. Sci Rep. 2020 Nov 20;10(1):20292. doi: 10.1038/s41598-020-77302-9.). Furthermore, we also compared both data of the gene expression and immunolabeling of the 3D HTM spheroid, and both results were quite similar, but some difference because spatial effects may be affected in terms of the immunolabeling. Similar results were also obtained from 3D spheroids derived from other source of cells including mouse and human adipocyte, human conjunctival fibroblasts and others. Therefore, we believe that our qPCR analysis is established methodology to estimate contents of specific molecule within the 3D spheroid.